# Transcriptome Analysis to Identify Responsive Genes under Sublethal Concentration of Bifenazate in the Diamondback Moth, *Plutella xylostella* (Linnaeus, 1758) (Lepidoptera: Plutellidae)

**DOI:** 10.3390/ijms232113173

**Published:** 2022-10-29

**Authors:** Qiuli Hou, Hanqiao Zhang, Jiani Zhu, Fang Liu

**Affiliations:** College of Horticulture and Plant Protection, Yangzhou University, Yangzhou 225009, China

**Keywords:** *Plutella xylostella*, bifenazate, sublethal concentrations, comparative transcriptome, gene expression changes

## Abstract

Bifenazate is a novel acaricide that has been widely used to control spider mites. Interestingly, we found bifenazate had a biological activity against the diamondback moth (*Plutella xylostella*), one of the most economically important pests on crucifer crops around the world. However, the molecular mechanisms underlying the response of *P. xylostella* to bifenazate treatment are not clear. In this study, we first estimated the LC_30_ dose of bifenazate for third-instar *P. xylostella* larvae. Then, in order to identify genes that respond to the treatment of this insecticide, the comparative transcriptome profiles were used to analyze the gene expression changes in *P. xylostella* larvae after exposure to LC_30_ of bifenazate. In total, 757 differentially expressed genes (DEGs) between bifenazate-treated and control *P. xylostella* larvae were identified, in which 526 and 231 genes were up-regulated and down-regulated, respectively. The further Kyoto Encyclopedia of Genes and Genomes (KEGG) analysis showed that the xenobiotics metabolisms pathway was significantly enriched, with ten detoxifying enzyme genes (four P450s, five glutathione S-transferases (GSTs), and one UDP-Glucuronosyltransferase (UGT)) were up-regulated, and their expression patterns were validated by qRT-PCR as well. Interestingly, the present results showed that 17 cuticular protein (CP) genes were also remarkably up-regulated, including 15 CPR family genes. Additionally, the oxidative phosphorylation pathway was found to be activated with eight mitochondrial genes up-regulated in bifenazate-treated larvae. In contrast, we found some genes that were involved in tyrosine metabolism and purine pathways were down-regulated, indicating these two pathways of bifenazate-exposed larvae were significantly inhibited. In conclusion, the present study would help us to better understand the molecular mechanisms of sublethal doses of bifenazate detoxification and action in *P. xylostella*.

## 1. Introduction

The diamondback moth, *Plutella xylostella* (Linnaeus, 1758) (Lepidoptera: Plutellidae), originally originated from the Mediterranean region, and it has become one of the most destructive agricultural pests for cruciferous crops around the world [1]. *P. xylostella* larvae mainly feed on the plant foliage, which could cause significant losses in the yield and quality of economically important cruciferous vegetables, such as cabbage, broccoli, cauliflower, etc. [2]. Even though some strategies have been developed to control the outbreak of *P. xylostella*, chemical insecticides are still one of the main measures to manage this insect [3]. Additionally, *P. xylostella* owned a high capacity for the rapid development of insecticide resistance, and due to the long-term and overuse of insecticides, it has evolved high resistance to almost all kinds of commercial insecticides, including chlorpyrifos, pyrethroids, carbamates, organophosphates, etc. [4,5]. Therefore, it is crucial to managing the insecticide resistance of *P. xylostella* to prevent further losses of these economically important crops.

Bifenazate [D2341, N0 -(4-methoxy-bipheny-3-yl) hydrazine carboxylic acid isopropyl ester] is a novel hydrazine carbazate acaricide, which was first discovered in 1990 by Uniroyal Chemical, and then commercialized in 1999 by Crompton Corporation [6]. As bifenazate exhibits quick knockdown via oral and contact activity, and this pesticide has been widely used to control various phytophagous leaf mites (*Tetranychus* spp., *Panonychus* spp. and *Oligonychus* spp.) [6,7]. Moreover, bifenazate owned excellent toxic activity on a wide range of currently used acaricide-resistant strains of spider mite, and thus it can be viewed as a resistance-breaking insecticide in the field [7,8].

To overcome the resistance problem of *P. xylostella*, many other groups of insecticides, such as macrocyclic lactones, neonicotinoids, diamides, etc., have been applied to control *P. xylostella* [9,10,11]. Interestingly, the present study showed that bifenazate had insecticidal activity to *P. xylostella* as well, which makes it an ideal potential insecticide for *P. xylostella* control. Additionally, understanding how the genes and pathways respond to insecticide stress may improve our understanding of the molecular mechanisms of chemical defense in *P. xylostella*. However, we still know little about the effects of bifenazate on the transcript levels of *P. xylostella*. In recent years, next-generation sequencing technologies have been dramatically developed, and RNA-sequencing (RNA-Seq) analysis has been proven to be a powerful and convenient method to identify differentially expressed genes (DEGs) of insects that treated with insecticides [12,13,14]. For example, previous studies have shown that exposure of *Bactrocera dorsalis* (Handel, 1912) (Diptera: Tephritidae) [15], *Anopheles sinensis* (Wiedemann, 1828) (Diptera: Culicidae) [16], *Drosophila melanogaster* (Meigen, 1830) (Diptera: Drosophilidae) [17], *Bombyx mori* (Linnaeus, 1758) (Lepidoptera: Bombycidae) [18] and *Spodoptera frugiperda* (Smith, 1797) (Lepidoptera: Noctuidae) [19] to sublethal concentrations of various insecticides caused the significant gene expression changes, including cathepsin family genes, heat shock protein genes, detoxification enzyme genes, etc.

In this study, we first conducted a bioassay to estimate the susceptibility of *P. xylostella* larvae to bifenazate, and the LC_30_ concentration of bifenazate was determined. Then, to evaluate the sublethal effects of bifenazate on the global gene expression changes of *P. xylostella*, the comparative transcriptome libraries were constructed between the bifenazate-treated *P. xylostella* larvae and the control. The DEGs were obtained, and their possible functions were predicted and enriched via functional annotation and enrichment analysis in different functional categories by searching against different databases. The DEGs that might be involved in detoxification responses were further confirmed by using RT-qPCR.

## 2. Results

### 2.1. Susceptibility of P. xylostella Exposed to Bifenazate

We evaluated the susceptibility of third-instar *P. xylostella* larvae to bifenazate to determine the LC_50_ and LC_30_ concentrations of bifenazate (Table 1). The estimated LC_50_ and LC_30_ values of bifenazate were 18.38 mg/L and 11.63 mg/L, respectively.

### 2.2. Overview of the RNA-Seq Data

In this paper, the RNA-Seq analysis was conducted on the insecticide treatment and control group in triplicate to explore the gene expression changes of *P. xylostella* that exposed LC_30_ of bifenazate. In total, 249,090,682 raw reads were obtained from six libraries of *P. xylostella*, which were subsequently submitted to the NCBI database (accession number: SRR20688928). Afterward, a strict quality control analysis for each *P. xylostella* sample was performed to confirm the qualification of this sequencing data. After eliminating the adaptor and low quality reads, 124,545,341 clean reads were retained, including 21,023,434, 20,899,860, and 20,764,706 and 19,706,331, 22,127,357, and 20,023,653 clean reads for each replicate of control group and bifenazate treatment, respectively (Table 2). The average of GC content, Q20, and Q30 of clean reads were 50.76%, 97.75%, and 93.79%, respectively (Table 2). Then, these clean reads were mapped to the reference genome, and mapped rates were 74.03%, 74.44%, and 73.80% and 76.81%, 75.65%, and 73.88% for each replicate of insecticide treatment and control group in *P. xylostella*, respectively (Table 2).

### 2.3. Analysis of Differentially Expressed Genes (DEGs)

In total, 14,712 genes were identified from six samples, and 13,624 genes could be successfully annotated in the *P. xylostella* genome, which is 92.60% of the total predicted genes (Appendix A). The remaining genes (1,088) that could not be successfully annotated in the *P. xylostella* genome were defined as novel genes. Importantly, we found significant changes at expression levels of these genes (|fold change| ≥ 2 with an FDR less than 0.05) between the insecticide treatment and control group in *P. xylostella*. Namely, 757 genes were identified as differentially expressed between the bifenazate treatment and control group of *P. xylostella* (Appendix A), among which 526 were up-regulated and 231 were down-regulated respectively (Figure 1A), and their detail expression patterns were shown in Figure 1B.

### 2.4. Gene Functional Annotations of DEGs

The present gene functional annotation of the DEGs showed that 723 genes were successfully identified in public databases, including Clusters of Orthologous Genes (COG) (220 genes), Gene Ontology (GO) (523 genes), Kyoto Encyclopedia of Genes and Genomes (KEGG) (475 genes), EuKaryotic Orthologous Groups (KOG) (361 genes), Non-redundant (Nr) (721 genes), Protein family (Pfam) (543 genes), Swiss-Prot (331 genes) and eggNOG (520 genes) (Appendix A). For further COG analysis, 220 DEGs were divided into 18 categories (Figure 2A), and the largest cluster was ‘Carbohydrate transport and metabolism’, followed by ‘Posttranslational modification, protein turnover, chaperones’, ‘Lipid transport and metabolism’, and ‘Amino acid transport and metabolism’. Moreover, we also found that 22, 14, and 12 genes were classified as ‘Secondary metabolites biosynthesis, transport and catabolism’, ‘Energy production and conversion’, and ‘Defense mechanisms’, respectively (Figure 2A). For the GO analysis, 523 DEGs were classified into 48 functional terms, which belonged to the three main groups, including biological process, cellular component, and molecular function (Figure 2B). For the biological process, the most representations were ‘cellular process’, ‘metabolic process’, and ‘single-organism process’. The cellular component group was mainly assigned to ‘membrane’, ‘membrane part’, ‘cell part’, and ‘cell’. The molecular function was mainly comprised of ‘catalytic activity’, ‘binding’, ‘transporter activity’, and ‘structural molecule activity’.

### 2.5. KEGG Enrichment Analysis of DEGs

Importantly, the subsequent KEGG pathway enrichment analysis showed that 248 DEGs were significantly enriched in the database, including 146 up-regulated and 102 down-regulated genes (Figure 3). For the up-regulated genes, the ‘Galactose metabolism’, ‘Drug metabolism-cytochrome P450′, ‘Metabolism of xenobiotics by cytochrome P450′ and ‘Amino sugar and nucleotide sugar metabolism’ pathways were significantly enriched (Figure 3A). Moreover, we also found 102 down-regulated genes were successfully enriched in KEGG pathways (Figure 3B), with six and ninegenes were involved in ‘Tyrosine metabolism’ and ‘Purine metabolism’ pathways (Table 3).

### 2.6. Changes in Expression Level of Detoxifying Genes

In this study, we found that ten genes involved in the detoxification metabolism pathway were up-regulated in *P. xylostella* after exposure to a sublethal dose of bifenazate (Appendix A), including three cytochrome P450s (P450s) in the CYP6 group (*CYP6AB5*, *CYP6AE22*, and *CYP6AW1*) and one P450s in CYP3 group (*CYP305B1*) (Figure 4); two glutathione S-transferases (GSTs) in epsilon family, two GSTs in delta family and one unclassified GST (Figure 5), and one UDP-glucuronosyltransferase (UGT) (Figure 6). Moreover, the results of the qRT-PCR analysis were consistent with the expression of ten up-regulated genes in RNA-seq data, which confirms the reliability of the DEG data (Figure 4B,C, Figure 5B,C and Figure 6B,C).

### 2.7. Changes in Expression Level of Cuticular Protein Genes (CPs)

Here, we found that 17 cuticular protein genes (CPs) were significantly up-regulated in the bifenazate treatment of *P. xylostella* (Figure 7). Further analysis showed that 15 CPs belonged to CPR families, including nine RR1 subfamily CPs (*RR1-8*, *RR1-9*, *RR1-28*, *RR1-36*, *RR1-38*, *RR1-39*, *RR1-41*, *RR1-48* and *RR1-49*), five RR2 subfamily CPs (*RR2-25*, *RR2-55*, *RR2-61*, *RR2-63* and *RR2-64*), one RR3 subfamily CP (*RR3-2*) (Appendix A). The remaining two CPs were divided into CPFL (CPF-like proteins) and 18aa families, respectively.

### 2.8. Changes in Expression Level of Mitochondrial Genes

In the present study, we found that the oxidative phosphorylation pathway was significantly enriched (Figure 8A), and eight genes encoding for enzymes that involved in mitochondrial functions were found to be up-regulated in *P. xylostella* after exposure to LC_30_ of bifenazate (Figure 8B). Moreover, these eight mitochondrial genes were classified into four groups, including two NADH dehydrogenase genes (*Ndufs5* and *Ndufa7*), one succinate dehydrogenase gene (*SDHA*), two cytochrome c oxidase (*COX6A* and *COX6B*), and three ATPase (*F-ATPase-β*, *F-ATPase-b* and *V-ATPase-D*) (Appendix A).

## 3. Discussion

Bifenazate is an efficient and safe pesticide, and thus it has become one the most frequently used insecticide for pest control worldwide, especially for spider mites [6,20]. To our knowledge, the present study was the first report to evaluate the susceptibility of *P. xylostella* larvae to bifenazate. Importantly, we found the toxicity activity of bifenazate to this destructive pest, suggesting it could be viewed as an alternative insecticide to control *P. xylostella*. Previous studies revealed that *P. xylostella* had evolved high resistance to various classes of insecticides, and sublethal selective pressure could promote the evolution of insecticide resistance in different insects [21,22,23]. Additionally, it is known that the identification of sublethal pesticide-responsive genes is required to help us better understand molecular mechanisms of insecticide action and the potential resistance of target insects [24].

With the development of next-generation high-throughput sequencing techniques, the transcriptome analysis has been considered to be an accurate and reliable means to study the characteristics of gene expression patterns under different abiotic stress in insect species, such as searching the critical factors associated with insecticide tolerance and resistance [25,26,27]. Recently, the developed comparative transcriptome has been used for *P. xylostella* to unveil its mechanisms of insecticide resistance evolution, developmental biology, and sex pheromone biosynthesis [3,28,29]. In this study, we exposed *P. xylostella* larvae to sublethal concentrations of bifenazate (LC_30_ = 11.63 mg/L) and then explored its transcriptional changes induced by bifenazate through the application of transcriptomic analysis. As expected, a large amount of DEGs was identified, including 526 up-regulated and 231 down-regulated genes in insecticide treatment. It has been proved that insects could evolve a powerful detoxification metabolism system to protect themselves against xenobiotics and pesticides, and the enhanced metabolism by the over-expression of detoxification genes (such as P450s, GSTs, and UGTs) played an important role in the physiological process of insects responding to insecticide exposure [30].

In this paper, the functional classification and enrichment analysis of DEGs were conducted, and the results showed that “xenobiotic detoxification metabolism” and “defense mechanisms” related pathways were significantly enriched, with ten detoxification enzyme genes up-regulated in bifenazate treatment of *P. xylostella*, including four P450 genes (three and one genes were divided into CYP6 and CYP3 group respectively), five GST genes (two and two genes belonged to epsilon and delta family) and one UGT gene. The P450s are a large group of detoxification enzymes that act as the terminal oxidases in the monooxygenase system to catalyze phase I reactions by directly metabolizing exogenous and endogenous substances [31,32]. It has been reported that P450 genes were up-regulated in response to xenobiotic pesticides in many insect species, particularly for CYP6 and CYP3 group genes [33]. Such as *CYP6B2*, *CYP6B6*, *CYP6B7*, *CYP6K1*, *CYP6B,* and *CYP321A* of *S. frugiperda* could be induced by different insecticides [19,34]. Eight P450s that included *CYP6CV5* and *CYP321F1* were over-expressed after exposure to a sublethal dose of chlorantraniliprole in *Chilo suppressalis* (Walker 1863) (Lepidoptera: Crambidae) [35]. It has also been shown that a *Cyp6a20* of *P. xylostella* was commonly over-transcribed after treatment with spinosad, chlorantraniliprole, cypermethrin, dinotefuran, and indoxacarb [4]. Similar to P450s, GSTs and UGTs are important detoxification enzymes in insects as well, which play critical roles in catalyzing phase II reactions by generating water-soluble products that could be easily excreted [36,37]. Many studies revealed that GSTs could be classified into six major groups, and epsilon and delta classes were mainly involved in insecticide metabolism and tolerance in insects [38,39]. For example, GSTs have been reported to be induced by various insecticides treatment, and delta and epsilon GSTs were closely associated with the detoxification of insecticide in *P. xylostella* [40]. For the UGTs, which have been confirmed to be involved in insecticide resistance, their expression levels could be induced by insecticides in insects [19,41]. Taken together, these results suggested that the up-regulated of P450s, GSTs and UGTs might offer the primary detoxification reactions against bifenazate exposure in *P. xylostella*, and future research is needed to prove this hypothesis.

The cuticle is the insect exoskeleton, which acts as the first barrier against diverse environmental stresses, such as insecticides, by altering the cuticle composition and structure [42]. CPs are major components of the insect cuticle, and the up-regulation of specific CPs was involved in insecticide penetration resistance [43]. CPs can be classified into more than 10 families, and CPR is the largest family protein that consists of a conserved chitin-binding domain, which contains three subfamilies, RR-1, RR-2, and RR-3. Previous studies have shown that the expression levels of numerous insect-specific CPs, particularly for CPR-type CPs, could be induced by insecticide exposure in *P. xylostella* [4,44], *D. melanogaster* [33], *S. frugiperda* [19], *C. suppressalis* [35] and *B. dorsalis* [45], or over-expressed in insecticide-resistant strains of *Cryptolestes ferrugineus* (Stephens, 1831) (Coleoptera: Laemophloeidae) [25], *Anopheles gambiae* (Giles, 1902) (Diptera: Culicidae) [46], *Cimex lectularius* (Linnaeus, 1758) (Hemiptera: Cimicidae) [47] and *Culex pipiens pallens* (Coquillett, 1898) (Diptera: Culicidae) [48]. Moreover, it has been proved that the over-expression of some CPR genes (*CPR124*, *CPR127*, and *CPR129*) was involved in pyrethroid resistance by enhancing the cuticle chitin layers of *A. gambiae* [49]. Knocking down of two CPR genes (*CpCPR63* and *CpCPR47*) by using RNA interference technology could lead to the increasing of pest mortality after deltamethrin exposure in *C. pipiens pallens* [48]. Similarly, we found 17 CPs (mainly belonging to the CPR family) were significantly strengthened in the bifenazate treatment, and according to previous studies described above, we implied that these up-regulated CP genes might play critical roles in the response and potentially the tolerance to bifenazate via reinforcing the chitin–protein matrix in the cuticle of *P. xylostella* [43,44].

Energy metabolism is a basic metabolic pathway, and one of its major important physiological functions is to supply energy sources for multifarious processes of organisms. Moreover, it has suggested that the changes in energy metabolism pathway represented a general feature of stress and insecticide exposure responses in different insects, including *D. melanogaster*, *Sitophilus zeamais* (Schoenherr, 1838) (Coleoptera: Curculionidae), *C. suppressalis* and *P. xylostella* [4,33,35,50]. Interestingly, the current study found ‘oxidative phosphorylation’ pathway was significantly activated, including two NADH dehydrogenase genes, one SDH gene, two COX genes, and three ATPase genes. The mitochondrial electron transport chain (mETC) consists of four protein complexes (I, II, III, and IV), which utilize a series of electron transfer reactions via oxidative phosphorylation to generate adenosine triphosphate (ATP) [51,52]. The complex I NADH dehydrogenase is the major electron entry site for the mETC, and thus it is critical for the production of mitochondrial ATP [53]. The complex II SDH has four subunits, which are involved in electron transfer from FADH2 (Flavine adenine dinucleotide, reduced) to ubiquinone, which is significant for ATP generation [54]. COX (complex IV enzyme) is a key site of cellular oxygen consumption, which is required for the production of aerobic energy in the form of ATP [55]. ATPase is a multi-subunit enzyme that can hydrolyze ATP to transport protons, which plays pivotal roles in maintaining normal biological processes and is essential for insect survival [56]. In the meantime, the KEGG analysis also found some up-regulated genes were significantly enriched in the galactose metabolism pathway. Galactose is an indispensable energy source for cellular metabolism, and it has been reported that galactose metabolism plays an important role in energy production and storage [57]. All these studies indicated that the up-regulation of mitochondrial genes might activate the energy metabolism pathway, which reflected the increase in energy consumption when *P. xylostella* suffered from bifenazate stress. Therefore, we inferred that the overall enhancement of energy generation and consumption might be required for *P. xylostella* to gain pesticide tolerance and refill the energy deficiency caused by exposure to sublethal concentrations of bifenazate.

In contrast to the over-expressed genes that followed bifenazate treatment in *P. xylostella*, this paper also showed six and nine down-regulated genes were significantly enriched in tyrosine metabolism and purine metabolism pathways, respectively. It has demonstrated that tyrosine metabolism is a complicated physiological process, which consists of a series of enzyme reactions that starts with tyrosine [58]. Moreover, tyrosine metabolism played an important role in the maintenance of normal insect development, and it has been confirmed that this pathway was essential for insect cuticle tanning (pigmentation and sclerotization) reproduction, survival, molting, immunity, and protecting the insect from an exogenous physical injury in different insect species [59,60,61]. As expected, previous studies have suggested that the repression of tyrosine metabolism by knocking down specific genes through RNA interference could significantly block the cuticle tanning and larval pupation process, impair melanization immune responses and eventually lead to high insect mortality [58,59,60,61,62,63]. Purines are abundant metabolic substrates that participate in different kinds of cellular processes, and they serve as the biosynthesis of DNA and RNA to confirm cell survival and proliferation [64]. Purine metabolism is a necessary response to oxidative stress, and the dysfunction and imbalance of this pathway could cause damaging reactive oxygen species (ROS) that lead to tremendous defects in physiological and pathological phenotypes [65,66]. For example, the inhibition of purine metabolism by knockout of the critical pathway genes could result in defects in the larval integument and male infertility of *B. mori* [67]. Additionally, many studies have shown that the exposure of insects to sublethal concentrations of insecticides could cause various physiological changes, such as adverse effects on insect development, growth, body weight, metamorphosis, and fecundity [35,68,69,70]. Here, based on our results, we speculated that the LC_30_ of bifenazate exposure might have sublethal toxic effects on the normal cuticle tanning, molting, pupation, or immunity of *P. xylostella* larvae via the restriction of tyrosine metabolism. The down-regulated genes that involved in purine metabolism pathway might cause translucent larval skin and toxic defects in cell survival of *P. xylostella* larvae after bifenazate exposure. As it should be, all these predictions needed to be further verified by future functional study.

## 4. Materials and Methods

### 4.1. Insect Sample and Insecticide

The laboratory colony population of *P. xylostella* was originally obtained from Jiangsu Province, People’s Republic of China, in 2017. All stages of *P. xylostella* were reared on radish seedlings [*Raphanuss ativus* (Linnaeus, 1753) (Rhoeadales: Brassicaceae)] at 27 ± 1 °C, relative humidity of 40–60%, and a photoperiod regime of 16:8 h light/dark without exposure to any pesticides. Technical grade bifenazate (98% active ingredient) was supplied by Shanghai yuan ye Bio-Technology Co., Ltd. (Shanghai, China).

### 4.2. Insecticide Bioassay

In this study, the leaf dip bioassay was used to test the susceptibility of *P. xylostella* larvae to bifenazate, and the detailed experimental procedures were based on the methods previously reported [71]. In brief, bifenazate was dissolved in acetone to make 1000 mg L^−1^ stock solution, and the distilled water that contained Triton X-100 (0.5 mg mL^−1^) was used to dilute the formulated insecticides into 7 serial dilutions. Cabbage [*Brassica oleracea* (Linnaeus, 1753) (Rhoeadales: Brassicaceae)] leaf discs (diameter = 6.5 cm) were cut and dipped in different solutions of the bifenazate for 30 s, and the control leaf discs were treated with distilled water that included 0.5% Triton X-100 and 1% acetone. After air drying for 1 h, the leaf discs were placed onto filter paper in plastic Petri dishes, and then the leaf discs were dried at room temperature for 2 h. For each concentration of bifenazate, twenty third-instar larvae were placed in each leaf disc with three replicates, which kept at 27 ± 1 °C and relative humidity of 40–60% with a photoperiod regime of 16:8 h light/dark. The insect mortality was assessed after 48 h, and the probit regression was constructed to calculate the LC_50_ and LC_30_ of *P. xylostella* via SPSS version 17.0 (SPSS, Chicago, IL, USA).

### 4.3. Insecticide Treatment and Total RNA Extraction

For the insecticide treatment, the third-instar *P. xylostella* larvae were collected and exposed to the LC_30_ bifenazate concentration with distilled water (containing 0.5% Triton X-100 and 1% acetone) as the control group. Here, three replicates were performed for each treatment, and each replicate contained thirty insects. Afterward, the living larvae were collected after 48 h exposure and immediately snap-frozen in liquid nitrogen for total RNA extraction. According to the manufacturer’s instructions, Trizol (Invitrogen, Carlsbad, CA, USA) was used for the RNA isolation. In order to eliminate contaminating genomic DNA, the total RNA was treated with DNase I (Promega, Madison, WI, USA). The RNA concentration was further quantified by using a NanoVue UV-Vis spectrophotometer (GE Healthcare Bio-Science, Uppsala, Sweden) to detect the absorbance at 260 nm. The absorbance ratios of OD260/280 and OD260/230 were applied to evaluate the RNA purity, and the integrity of RNA was confirmed by using the 1% agarose gel electrophoresis.

### 4.4. Library Construction and Sequencing

A total amount of 1 μg RNA per sample was used as input material for the RNA sample preparations. The cDNA libraries were generated by using NEBNext UltraTM RNA Library Prep Kit for Illumina (New England Biolabs, Ipswich, USA) following manufacturer’s recommendations. In brief, the total RNA was purified to generate mRNA by using poly-T oligo-attached magnetic beads. The obtained mRNAs were further broken into pieces by using divalent cations under elevated temperature in NEBNext First Strand Synthesis Reaction Buffer (5X). Then, the first-strand cDNA was synthesized via the random hexamer primer and M-MuLV Reverse Transcriptase, and the second-strand cDNA was synthesized by using RNase H and DNA polymerase I.

The cDNA library fragments were purified with AMPure XP system (Beckman Coulter, Beverly, CA, USA) to select fragments of preferentially 240 bp in length. Then, 3 μL USER Enzyme (New England Biolabs, Ipswich, USA) was used for size-selected and adaptor-ligated cDNA at 37 °C for 15 min, followed by 5 min at 95 °C before PCR. The PCR was performed to obtain the cDNA library, and its quality was assessed on the Agilent Bioanalyzer 2100 system. Lastly, the resulting cDNA library was sequenced on an Illumina platform (Illumina NovaSeq 6000), and paired-end reads were generated.

### 4.5. Analysis of RNA-Seq Data

In this study, the raw reads of fastq format were firstly processed through in-house perl scripts to obtain the high-quality clean reads that removed the reads containing adapter, reads containing ploy-N, and low-quality reads (base quality less than or equal to 5 is more than 20%) from raw data. Moreover, the GC-content, Q20, and Q30 of the clean data were calculated, and the downstream analyses were based on these high-quality clean data. These clean reads were then mapped to the *P. xylostella* genome assembly (NCBI Assembly: GCA_000330985.1) by using HISAT2 [72], and only reads with a perfect match or one mismatch were further analyzed and annotated based on the reference genome.

The quantification of gene expression levels was estimated by fragments per kilobase of transcript per million fragments mapped (FPKM). The Pearson correlation was used to confirm the faithful replicates of RNA-seq results via analyzing the FPKM between the biological replications [73,74]. The differential gene expression between the treatment and control groups was determined by using DEGseq package with a model based on the negative binomial distribution [75]. The false discovery rate (FDR) was calculated to adjust the *p*-value in the multiple testing by using Benjamini and Hochberg’s approaches. The genes that had a |fold change| ≥ 1.5 and FDR < 0.05 were defined as the differentially expressed genes (DEGs).

### 4.6. Gene Functional Annotations of DEGs

The obtained DEGs were blasted against the Clusters of Orthologous Groups (COG), Gene Ontology (GO), Kyoto Encyclopedia of Genes and Genomes (KEGG), EuKaryotic Orthologous Groups (KOG), Non-redundant (Nr), Protein family (Pfam), Swiss-Prot and eggNOG databases to obtain genes annotation information and functional classification with an *E*-value < 10^−5^.

### 4.7. KEGG Pathway Enrichment Analysis of DEGs

KEGG [76] is a database resource for understanding high-level functions and utilities of the biological system, such as the cell, the organism, and the ecosystem, from molecular-level information, especially large-scale molecular datasets generated by genome sequencing and other high-throughput experimental technologies (http://www.genome.jp/kegg/, accessed on 28 July 2022). In this study, the KOBAS (http://kobas.cbi.pku.edu.cn/expression.php, accessed on 28 July 2022) software was performed for the KEGG enrichment analysis of DEGs [77]. Namely, all DEGs were mapped into KEGG terms in the database, and the enriched terms were annotated by using Phyper (https://en.wikipedia.org/wiki/Hypergeometric_distribution, accessed on 20 September 2022) based on Hypergeometric test. The significant levels of KEGG terms were corrected by q-value with a threshold (q-value ≤ 0.05) by Bonferroni.

### 4.8. Quantitative Real-Time Polymerase Chain Reaction (qRT-PCR)

In this study, the DEGs that encoded detoxification-related enzymes were selected for the qRT-PCR analysis to validate the reliability of the present sequencing, and the primers were listed in Appendix A. As described above, we used the TRIzol reagent to extract the total RNA from the pesticide treatment (LC_30_) and control group of *P. xylostella*. Then, the total RNAs were treated with DNase I (Promega, Madison, WI, United States) to clean out the genomic DNA contamination, and the reverse transcriptase (Takara) and oligo d (T) primer (Takara) were used to synthesize the first-strand cDNAs. Here, the 10 μL reaction system was applied for the qRT-PCR analysis, which included 0.5 μL of cDNA samples, 0.3 μL of each primer (10 mM), 5 μL of GoTaq qRT-PCR Master Mix (Promega) and 3.9 μL of nuclease-free water. The detailed procedure of qRT-PCR experiment was an initial denaturation at 95 °C for 2 min, then 40 cycles with 95 °C for 15 s and 60 °C for 30 s by using the CFX 96 Touch Real-Time PCR detection system (USA, Bio-Rad). Meanwhile, the melting curve analysis was conducted for all reactions from 60 °C to 95 °C to confirm the consistency and specificity of PCR products. We used the RPL32 gene (GenBank accession no. AB180441) as the reference gene [44,78], and the 2^ΔΔCT^ method [79] was performed to calculate the relative expression changes of selected DEGs. Here, the qRT-PCR experiments were carried out with three biological replicates.

## 5. Conclusions

In summary, the susceptibility of *P. xylostella* larvae to bifenazate was evaluated, and then the sublethal concentration (LC_30_) of insecticide was calculated. Subsequently, we studied the sublethal effects of bifenazate on the global gene expressions of *P. xylostella* using the comparative transcriptome. Totally, 757 DEGs were identified with 526 and 231 genes overexpressed and down-regulated, respectively. The further functional annotation analysis suggested that detoxification enzyme genes (P450s, GSTs and UGTs), CP genes, and mitochondrial genes were involved in response to against sublethal dose of bifenazate stress and defense. In addition, we also inferred that the inhibition of tyrosine and purine metabolism pathways might contribute to the adverse effect on the survival or development of *P. xylostella*. In a word, the present extensive-expression data provided us the valuable insights into molecular mechanisms of *P. xylostella* responses and adaptions to sublethal bifenazate exposure, and future functional analysis is required to study the roles of these specific genes and pathways in this process.

## Figures and Tables

**Figure 1 ijms-23-13173-f001:**
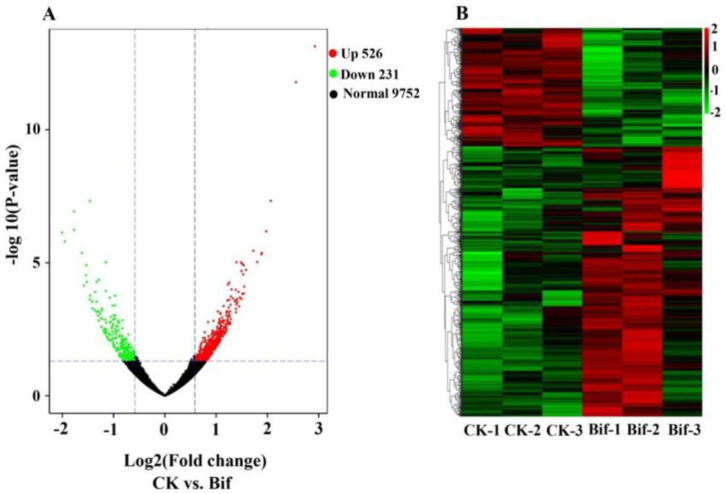
(**A**) Volcano plot of differentially expressed genes (DEGs, |fold change| ≥ 1.5 and FDR < 0.05), and the up-regulated genes were represented by a red dot and down-regulated genes by a green dot. (**B**) Expression profiles (FPKM value) of DEGs. *X*-axis: sample name, *Y*-axis: DEGs. The colors indicated the expression levels of DEGs, and red indicated high expression and green indicated low expression.

**Figure 2 ijms-23-13173-f002:**
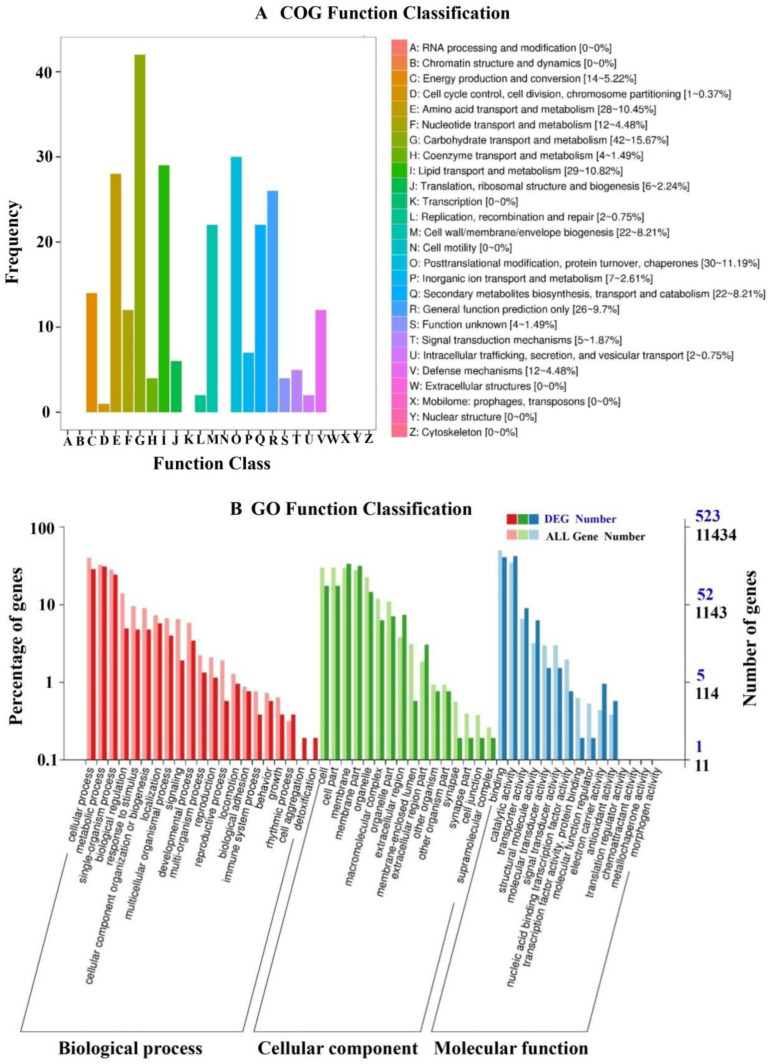
(**A**) COG function classifications of the differentially expressed genes (DEGs). (**B**) GO function classifications of DEGs. The *X*-axis represents names of function classification, and the *Y*-axis corresponds to the number of genes in each function classification.

**Figure 3 ijms-23-13173-f003:**
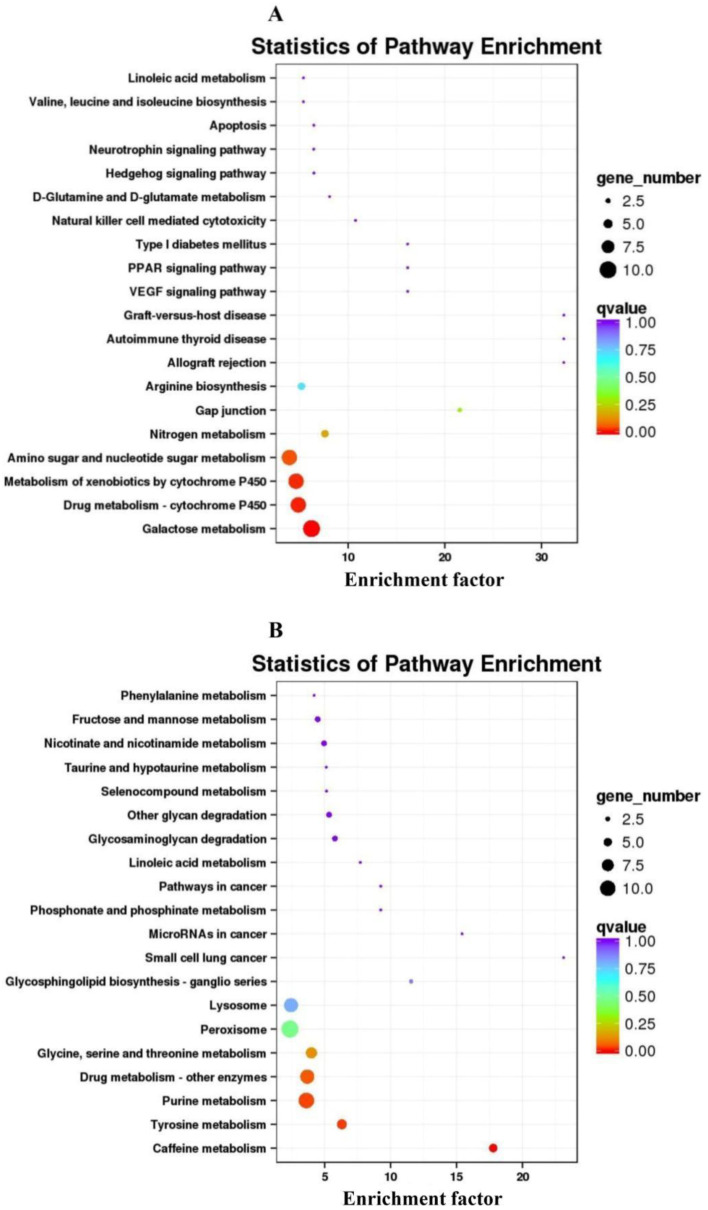
(**A**) KEGG significant enrichment analysis for the up-regulated genes. (**B**) KEGG significant enrichment analysis for the down-regulated genes. The significant enriched pathways were shown with their q-value (color), enrichment factor (*X*-axis), and a number of involved differentially expressed genes (size of circles).

**Figure 4 ijms-23-13173-f004:**
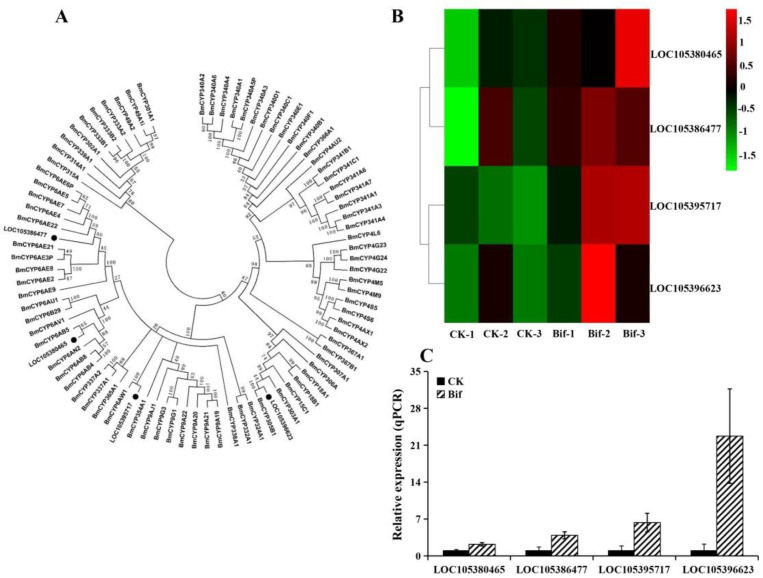
(**A**) Phylogenetic relationships of P450s from *Plutella xylostella* (black circle) and *Bombyx mori*. (**B**) Expression profiles (FPKM value) of four P450s in *P. xylostella*. *X*-axis: sample name, *Y*-axis: P450s. The colors indicated the expression levels of P450s, and red indicated high expression and green indicated low expression. (**C**) Expression profiles of four P450s in *P. xylostella* by qRT-PCR.

**Figure 5 ijms-23-13173-f005:**
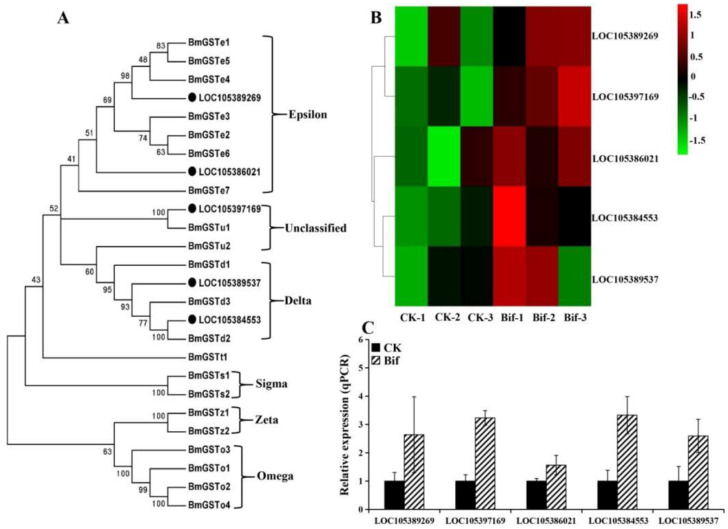
(**A**) Phylogenetic relationships of GSTs from *Plutella xylostella* (black circle) and *Bombyx mori*. (**B**) Expression profiles (FPKM value) of five GSTs in *P. xylostella*. *X*-axis: sample name, *Y*-axis: GSTs. The colors indicated the expression levels of GSTs, and red indicated high expression and green indicated low expression. (**C**) Expression profiles of five GSTs in *P. xylostella* by qRT-PCR.

**Figure 6 ijms-23-13173-f006:**
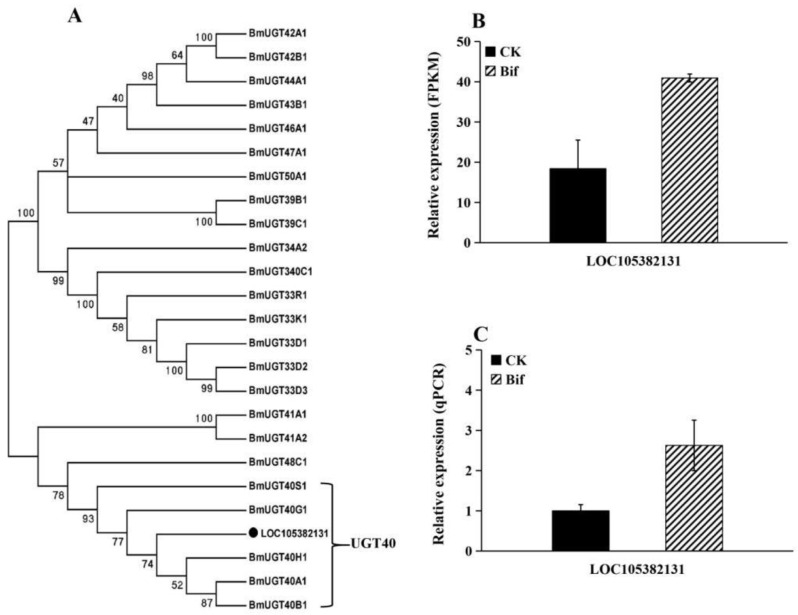
(**A**) Phylogenetic relationships of UGTs from *Plutella xylostella* (black circle) and *Bombyx mori*. (**B**) Expression profiles (FPKM value) of a UGT in *P. xylostella*. (**C**) Expression profiles of a UGT in *P. xylostella* by qRT-PCR.

**Figure 7 ijms-23-13173-f007:**
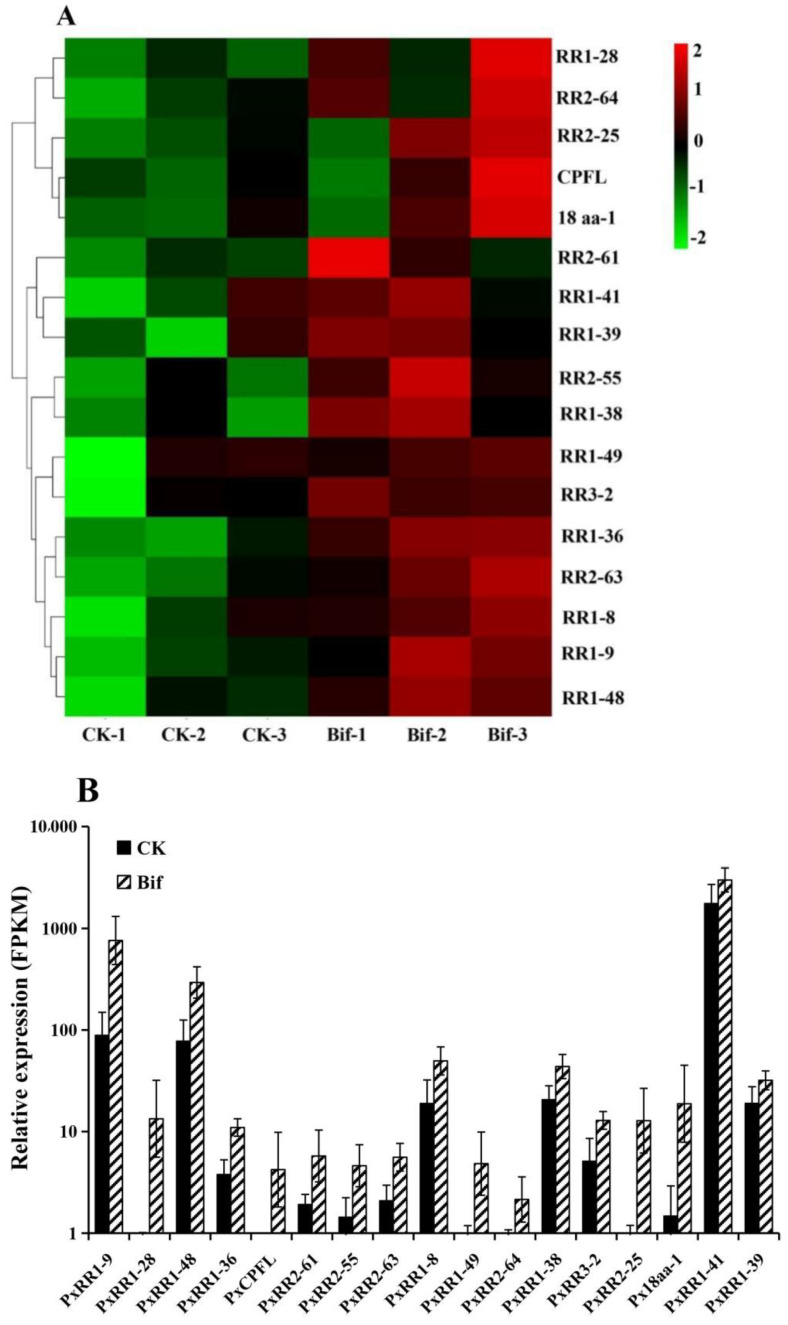
The expression levels (FPKM value) of 17 cuticular protein (CP) genes in *Plutella xylostella* treated with bifenazate at concentrations of LC_30_. (**A**) Heatmap of 17 CPs expression levels. *X*-axis: sample name, *Y*-axis: CPs. The colors indicated the expression levels of CPs, and red indicated high expression and green indicated low expression. (**B**) Histogram analysis of 17 CPs expression levels.

**Figure 8 ijms-23-13173-f008:**
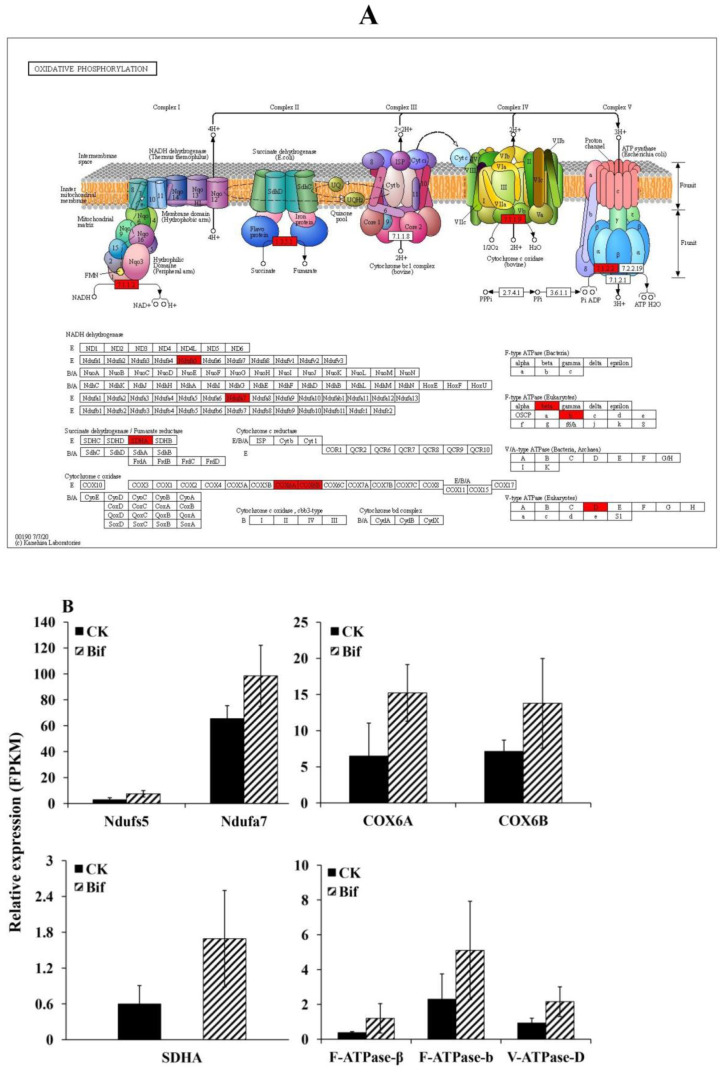
(**A**) The KEGG pathway of the oxidative phosphorylation pathway responds to bifenazate treatment in *Plutella xylostella*, and genes highlighted in green are enriched and down-regulated. (**B**) The expression levels (FPKM value) of eight mitochondrial genes in *P. xylostella* that treated with bifenazate at concentrations of LC_30_.

**Table 1 ijms-23-13173-t001:** Susceptibility of *Plutella xylostella* larvae to bifenazate.

Regression Equation	LC_50_ (95% CI) (mg/L)	LC_30_ (95% CI) (mg/L)	χ^2^
y = −2.97 + 2.35x	18.38 (3.55–32.14)	11.63 (1.97–17.36)	10.73

LC_50_ = Lethal Concentration for 50%, LC_30_ = Lethal Concentration for 30%, CI = Confidence Interval.

**Table 2 ijms-23-13173-t002:** Data output quality and mapping rates for the examined samples of *Plutella xylostella*.

Samples	Clean Reads	GC Content	Q20 (%)	Q30 (%)	Genome
CK-1	21,023,434	49.79%	97.52	93.22	74.03
CK-2	20,899,860	50.33%	97.80	93.89	74.44
CK-3	20,764,706	51.39%	97.71	93.77	73.80
Bif-1	19,706,331	50.53%	97.64	93.47	76.81
Bif-2	22,127,357	52.05%	98.06	94.53	75.65
Bif-3	20,023,653	50.48%	97.75	93.85	73.88

Clean reads ratio (%) = Total clean reads/Total raw reads.

**Table 3 ijms-23-13173-t003:** The down-regulated genes that involved in tyrosine metabolism and purine metabolism pathways were identified in bifenazate treatment of *Plutella xylostella*.

Gene ID	*p*-Value	Log2FC	KEGG Pathway ID	Nr Annotation
LOC105381553	4.24 × 10^−6^	−1.62	ko00350	juvenile hormone-suppressible protein
LOC105382593	0.023	−0.80	ko00350	aromatic-L-amino-acid decarboxylase
LOC105386870	0.016	−0.92	ko00350	prophenoloxidase
LOC105388111	0.0028	−1.04	ko00350	acidic juvenile hormone-suppressible protein
LOC105388114	0.014	−0.84	ko00350	arylphorin subunit beta-like precursor
LOC119690367	5.82 × 10^−7^	−1.77	ko00350	juvenile hormone-suppressible protein
LOC105381730	0.030	−0.65	ko00230	adenosine deaminase CECR1-A-like
LOC105382860	0.027	−0.60	ko00230	xanthine dehydrogenase
LOC105383429	0.025	−0.64	ko00230	apyrase-like
LOC105383538	0.0017	−1.075	ko00230	indole-3-acetaldehyde oxidase-like
LOC105384208	0.022	−0.69	ko00230	xanthine dehydrogenase
LOC105386323	0.024	−0.86	ko00230	uricase
LOC105393481	0.043	−0.59	ko00230	purine nucleoside phosphorylase-like
LOC105393606	0.041	−0.66	ko00230	phosphoribosylformylglycinamidine synthase
LOC105398645	0.0075	−0.83	ko00230	amidophosphoribosyltransferase-like

KEGG pathway ID ko00350 and ko00230 represented ‘Tyrosine metabolism’ and ‘Purine metabolism’ pathway, respectively.

## Data Availability

All data are included in the text and Appendix A.

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
