# Peer review of "Transcriptome Analysis to Identify Responsive Genes under Sublethal Concentration of Bifenazate in the Diamondback Moth, Plutella xylostella (Linnaeus, 1758) (Lepidoptera: Plutellidae)"

_ijms, 2022, doi:10.3390/ijms232113173_

Round 1

Reviewer 1 Report

Line 4

Change “Plutella xylostella” for “Plutella xylostella (Linnaeus, 1758) (Lepidoptera: Plutellidae)”

Line 19

Change “757 DEGs (differentially expressed genes)” for “757 differentially expressed genes (DEGs)

Line 21

Change “KEGG” for “Kyoto Encyclopedia of Genes and Genomes (KEGG)”

Line 23

Change “GSTs” for “glutathione S-transferases (GSTs)”

Change “UGT” for “UDP-Glucuronosyltransferase (UGT)”

Line 36

Change “Plutella xylostella (L.) (Lepidoptera: Plutellidae)” for “Plutella xylostella (Linnaeus, 1758) (Lepidoptera: Plutellidae)”

Lines 49 and 50

Change “Bifenazate (D2341, N0 -(4-methoxy-bipheny-3-yl) hydrazine carboxylic acid isopropyl ester)” for ““Bifenazate [D2341, N0 -(4-methoxy-bipheny-3-yl) hydrazine carboxylic acid isopropyl ester]”

Line 54

Change “(Tetranychus spp., Panonychus spp. and Oligonychus spp.)” for “(Tetranychus spp., Panonychus spp. and Oligonychus spp.)”

Lines 69 and 70

Change “Bactrocera dorsalis [15], Anopheles sinensis [16], Drosophila melanogaster [17], Bombyx mori [18] and Spodoptera frugiperda [19]” for “Bactrocera dorsalis (Handel, 1912) (Diptera: Tephritidae) [15], Anopheles sinensis Wiedemann, 1828 (Diptera: Culicidae) [16], Drosophila melanogaster Meigen, 1830 (Diptera: Drosophilidae) [17], Bombyx mori Linnaeus, 1758 (Lepidoptera: Bombycidae) [18] and Spodoptera frugiperda (Smith, 1797) (Lepidoptera: Noctuidae) [19]”

Line 109

(Table S2) - Change the numbering of supplementary tables so that they appear in ascending numerical order.

Line 125 and 126

including COG (220 genes), GO (523 genes), KEGG (475 genes), KOG (361 genes), NR (721 genes), Pfam (543 genes), Swiss-Prot (331 genes) and eggNOG (520 genes) (Fig. S1).

Write all names in full the first time they appear on the manuscript. For example: change “KEGG (475 genes)” for “Kyoto Encyclopedia of Genes and Genomes (KEGG) (475 genes)”

Page 141 onwards

All images need to be reviewed. Figures 2 and 3, for example, need to have better resolution and larger size. It is not possible to read what is written. I suggest that the authors separate A and B and present the images with a focus and size that allows reading.

Line 249

Change “Chilo suppressalis” for ““Chilo suppressalis (Walker 1863) (Lepidoptera: Crambidae)”

Lines 270-278, 287, 338 and 347

Include full scientific names when first cited (as I am doing here in the review)

Author Response

Reviewer 1

Line 4

Change “Plutella xylostella” for “Plutella xylostella (Linnaeus, 1758) (Lepidoptera: Plutellidae)”

Reply: It has been changed.

Line 19

Change “757 DEGs (differentially expressed genes)” for “757 differentially expressed genes (DEGs)

Reply: It has been changed.

Line 21

Change “KEGG” for “Kyoto Encyclopedia of Genes and Genomes (KEGG)”

Reply: It has been changed.

Line 23

Change “GSTs” for “glutathione S-transferases (GSTs)”

Change “UGT” for “UDP-Glucuronosyltransferase (UGT)”

Reply: They have been changed.

Line 36

Change “Plutella xylostella (L.) (Lepidoptera: Plutellidae)” for “Plutella xylostella (Linnaeus, 1758) (Lepidoptera: Plutellidae)”

Reply: It has been changed.

Lines 49 and 50

Change “Bifenazate (D2341, N0 -(4-methoxy-bipheny-3-yl) hydrazine carboxylic acid isopropyl ester)” for ““Bifenazate [D2341, N0 -(4-methoxy-bipheny-3-yl) hydrazine carboxylic acid isopropyl ester]”

Reply: It has been changed.

Line 54

Change “(Tetranychus spp., Panonychus spp. and Oligonychus spp.)” for “(Tetranychus spp., Panonychus spp. and Oligonychus spp.)”

Reply: It has been changed.

Lines 69 and 70

Change “Bactrocera dorsalis [15], Anopheles sinensis [16], Drosophila melanogaster [17], Bombyx mori [18] and Spodoptera frugiperda [19]” for “Bactrocera dorsalis (Handel, 1912) (Diptera: Tephritidae) [15], Anopheles sinensis Wiedemann, 1828 (Diptera: Culicidae) [16], Drosophila melanogaster Meigen, 1830 (Diptera: Drosophilidae) [17], Bombyx mori Linnaeus, 1758 (Lepidoptera: Bombycidae) [18] and Spodoptera frugiperda (Smith, 1797) (Lepidoptera: Noctuidae) [19]”

Reply: It has been changed.

Line 109

(Table S2) - Change the numbering of supplementary tables so that they appear in ascending numerical order.

Reply: It has been adjusted.

Line 125 and 126

including COG (220 genes), GO (523 genes), KEGG (475 genes), KOG (361 genes), NR (721 genes), Pfam (543 genes), Swiss-Prot (331 genes) and eggNOG (520 genes) (Fig. S1).

Write all names in full the first time they appear on the manuscript. For example: change “KEGG (475 genes)” for “Kyoto Encyclopedia of Genes and Genomes (KEGG) (475 genes)”

Reply: According to the reviewer’s suggestions, the full names have been provided.

Page 141 onwards

All images need to be reviewed. Figures 2 and 3, for example, need to have better resolution and larger size. It is not possible to read what is written. I suggest that the authors separate A and B and present the images with a focus and size that allows reading.

Reply: According to the reviewer’s suggestions, the images have been reviewed and adjusted.

Line 249

Change “Chilo suppressalis” for ““Chilo suppressalis (Walker 1863) (Lepidoptera: Crambidae)”

Reply: It has been adjusted.

Lines 270-278, 287, 338 and 347

Include full scientific names when first cited (as I am doing here in the review)

Reply: It has been adjusted.

Reviewer 2 Report

I congratulate the authors on their work and manuscript. Manuscript IJMS-1930770 "Transcriptome analysis to identify responsive genes under sublethal concentration of bifenazate in the diamondback moth, Plutella xylostella" by Hou and collaborators is objective and presents the first transcriptome investigation of this important insect treated with the novel acaricide Bifenazate. First, the LC50 and LC30 concentrations were determined, and then RNAseq analysis was performed. The experimental design is straightforward and data on selected genes was further validated by RT-qPCR. The authors were able to identify an interesting set of genes that help to understand the effects of sublethal treatments. Although the text is well written, there are a few necessary adjustments to bring the manuscript to MDPI standards. I listed some pages below to guide corrections but I recommend a complete review by a professional.

Pages for English adjustments: 13, 53, 107, 151, 186, 210, 260-264, 329, 344, 445.

Other minor suggestions:
Line 212: The authors can use "To our knowledge" instead of "As far as I know".

In figure 3A, instead of "Rich factor", the authors can use "Enrichment factor".

A suggestion for Table 3 is to remove the "Regulated" column, reduce the space between all columns and include one with "Annotation" to provide a functional clue of each gene. If necessary to create more space, the authors can show just the ko number in the "KEGG pathway" column and write Tyrosine and Purine metabolism as footnotes.

It would contribute to the paper if the authors included a figure and description of insects after bifenazate treatment.

Author Response

Reviewer 2

I congratulate the authors on their work and manuscript. Manuscript IJMS-1930770 "Transcriptome analysis to identify responsive genes under sublethal concentration of bifenazate in the diamondback moth, Plutella xylostella" by Hou and collaborators is objective and presents the first transcriptome investigation of this important insect treated with the novel acaricide Bifenazate. First, the LC50 and LC30 concentrations were determined, and then RNAseq analysis was performed. The experimental design is straightforward and data on selected genes was further validated by RT-qPCR. The authors were able to identify an interesting set of genes that help to understand the effects of sublethal treatments. Although the text is well written, there are a few necessary adjustments to bring the manuscript to MDPI standards. I listed some pages below to guide corrections but I recommend a complete review by a professional.

Reply: I am grateful that the reviewer likes this paper, and according to the reviewer’s suggestion, the necessary adjustments have been conducted.

Pages for English adjustments: 13, 53, 107, 151, 186, 210, 260-264, 329, 344, 445.

Reply: The English adjustments have been completed.

Other minor suggestions:

Line 212: The authors can use "To our knowledge" instead of "As far as I know".

Reply: The English adjustments have been completed.

In figure 3A, instead of "Rich factor", the authors can use "Enrichment factor".

Reply: This figure has been adjusted.

A suggestion for Table 3 is to remove the "Regulated" column, reduce the space between all columns and include one with "Annotation" to provide a functional clue of each gene. If necessary to create more space, the authors can show just the ko number in the "KEGG pathway" column and write Tyrosine and Purine metabolism as footnotes.

Reply: According to the reviewer’s suggestion, the table 3 has been adjusted.

It would contribute to the paper if the authors included a figure and description of insects after bifenazate treatment.

Reply: I agreed with reviewer's viewpoint that a figure and description of insects after bifenazate treatment could contribute to the paper. Besides, the present experiment was mainly conducted to identify genes that responding to sublethal treatments of bifenazate in Plutella xylostella. In future, according to gene expression changes in P. xylostella larvae after exposure to LC30 of bifenazate, the study will be conducted to predict and observe its adverse effects on insect development (including figures and description of insects after bifenazate treatment). Then, the RNA interference (RNAi) technology will be used to verify the genes that might be involved in bifenazate insecticidal mechanisms in P. xylostella.

Reviewer 3 Report

The manuscript attempted to identify putative genes responding to sublethal doses of an acaricide. While molecular information was yielded by RNASeq and bioinfo analysis to predict the genes likely associated with the mode of action of Bifenazate to P. xylostella, the manuscript lacked supporting information that would associate molecular response to the manifested sub-lethal effects.

Major Comments:

1. Susceptibility studies of Bifenazate compared with other insecticides were not presented, but authors mentioned of its high activity. 2. Sub-lethal effects manifested not only as mortality but for example as reductions in life span, development rates, fecundity, deformities, changes in behavior, feeding, were not recorded and/or shown. These could have been shown to corroborate RNAseq results. Please see Lines 28-29 claiming that normal growth and development of bifenazate-exposed larvae were significantly inhibited.

3. In relation to #2. Below are comments in relation to the authors’ claim of the involvement of tyrosine and purine metabolism pathway genes that the authors may want to address: In insects, regulation of amino acid metabolism, particularly tyrosine, is crucial for oogenesis, and survival. Insects just like other animals can obtain tyrosine from the hydroxylation of phenylalanine or from the hydrolysis of food proteins. It is also known that insects rely heavily on tyrosine for cuticle hardening and for the melanization of pathogens as a key component of the insect immune system. Further, tyrosine metabolism enzymes have also been shown to affect negatively reproductive fitness, and moulting of insects. In recognition of the above, the speculation that Bifenazate acts on tyrosine metabolism pathway is not supported by the DEG results. First, 17 cuticular proteins were upregulated implying reinforcement of cuticular-protein matrices; thus hardening. But tyrosine is down-regulated?

Second, downregulation of purine might also lead to changes in of the larval skin that would cause it to become affected by light. Such is the case of silkworms with mutations in several genes along the purine metabolism pathway; causing

translucent larval skin implying changes in uric acid accumulation and protect them from light. The downregulation of purine could not be associated with cuticle hardening then in the case of this study. 4. Illumina platform used was not mentioned. 5. I am not sure if this was an honest mistake, but I could not find the RNASeq data in NCBI. What I found is the SRA for the flat lined beetle, Cryptolestes ferrugineus.

Author Response

Dear reviewer

Thank you very much for your regarding to our manuscript ID ijms-1930770 entitled "Transcriptome analysis to identify responsive genes under sublethal concentration of bifenazate in the diamondback moth, Plutella xylostella". The comments were all valuable and helpful. We have incorporated all suggestions and comments of the reviewer.

With best regards,

All authors

Major Comments:

  1. Susceptibility studies of Bifenazate compared with other insecticides were not presented, but authors mentioned of its high activity.

Reply: I agreed with reviewer that the description of high activity was unsuitable, when the comparative experiments were not conducted. Therefore, those inappropriate language descriptions have been adjusted.

  1. Sub-lethal effects manifested not only as mortality but for example as reductions in life span, development rates, fecundity, deformities, changes in behavior, feeding, were not recorded and/or shown. These could have been shown to corroborate RNAseq results. Please see Lines 28-29 claiming that normal growth and development of bifenazate-exposed larvae were significantly inhibited.

Reply: As the reviewer said that the sub-lethal effects on insect phenotype included reductions in life span, development rates, fecundity, deformities, changes in behavior, feeding etc. As for the present paper, the experiment was mainly conducted to identify genes that responding to sublethal treatments of bifenazate in Plutella xylostella. In future, according to gene expression changes in P. xylostella larvae after exposure to LC30 of bifenazate, the study will be conducted to predict and observe its adverse effects on insect development. Then, the RNA interference (RNAi) technology will be used to verify the genes that might be involved in bifenazate insecticidal mechanisms in P. xylostella.

Moreover, in this study, we found some down-regulated genes were invovled in tyrosine metabolism and purine pathways, and these pathways were crucial for insects. Therefore, according to the results of RNA-seq analysis, we speculated that the normal growth and development of bifenazate-exposed larvae might be significantly inhibited. In order to make this statement more accurate, the Lines 28-29 has been adjusted accordingly.

  1. In relation to #2. Below are comments in relation to the authors’ claim of the involvement of tyrosine and purine metabolism pathway genes that the authors may want to address: In insects, regulation of amino acid metabolism, particularly tyrosine, is crucial for oogenesis, and survival. Insects just like other animals can obtain tyrosine from the hydroxylation of phenylalanine or from the hydrolysis of food proteins. It is also known that insects rely heavily on tyrosine for cuticle hardening and for the melanization of pathogens as a key component of the insect immune system. Further, tyrosine metabolism enzymes have also been shown to affect negatively reproductive fitness, and moulting of insects.

Reply: As the reviewer said that tyrosine metabolism played the important roles in the maintenance of insect normal development, and which has been confirmed that this pathway was essential for insect cuticle tanning (pigmentation and sclerotization) reproduction, survival, molting, immunity and protecting the insect from exogenous physical injury in different insect species. Therefore, we speculated that the down-regulated genes that involved in tyrosine metabolism pathway might imply the sublethal toxic effects on the normal cuticle tanning, molting, pupation, and immunity of P. xylostella larvae after the LC30 of bifenazate exposure. Moreover, all these information described above have been adjusted in this paper.

In recognition of the above, the speculation that Bifenazate acts on tyrosine metabolism pathway is not supported by the DEG results. First, 17 cuticular proteins were upregulated implying reinforcement of cuticular-protein matrices; thus hardening. But tyrosine is down-regulated?

Reply: Cuticle proteins (CPs) are major components of the insect cuticle, which interact with each other and with chitin to form a stabilized cuticular structure that maintains the elasticity and other physical properties of the insect exoskeleton. It has been reported the up-regulation of specific CPs were involved to against diverse environmental stresses, such as insecticides. Previous studies have showed that the expression levels of numerous insect specific CPs, particularly for CPR type (consisted of a conserved chitin-binding domain) CPs, could be induced by insecticide exposure in P. xylostella. As expected, we found 17 CPs (mainly belonging to CPR family) were significantly strengthened in the bifenazate treatment, and we implied that these up-regulated CP genes might play the critical roles in the response and potentially the tolerance to bifenazate via reinforcing the chitin-protein matrix in the cuticle of P. xylostella.

However, the reviewer thought that the speculation that bifenazate acts on tyrosine metabolism pathway is not supported by the DEG results, because of 17 cuticular proteins were up-regulated. But I think that such results are reasonable, and the opposite expression pattern of CPs and tyrosine metabolism pathway genes implied a compensation mechanism. Namely, the bifenazate exposure might cause the defects in larval cuticle tanning of P. xylostella, and meanwhile the up-regulated CPs might help balance such defects by the reinforcement of chitin-protein matrix in the cuticle.

Second, downregulation of purine might also lead to changes in of the larval skin that would cause it to become affected by light. Such is the case of silkworms with mutations in several genes along the purine metabolism pathway; causing translucent larval skin implying changes in uric acid accumulation and protect them from light. The downregulation of purine could not be associated with cuticle hardening then in the case of this study.

Reply: Yes, in this study, we also thought that the down-regulated genes that involved in purine pathway were not associated with cuticle hardening. Maybe some of my inappropriate descriptions have misled the reviewer, and the corresponding sentences have been adjusted.

  1. Illumina platform used was not mentioned.

Reply: In this study, Illumina NovaSeq 6000 platform was used for sequencing, and we have added this information into this paper.

  1. I am not sure if this was an honest mistake, but I could not find the RNASeq data in NCBI. What I found is the SRA for the flat lined beetle, Cryptolestes ferrugineus.

Reply: I am so sorry for this unintentional mistake, the corrected NCBI accession number is SRR20688928, which has been provided in this paper.

Round 2

Reviewer 3 Report

Authors provided clarifications regarding technical concerns, data deposition and  interpretation. The manuscript can now be published. Minor text editing is suggested.